# Understanding Fear after an Anterior Cruciate Ligament Injury: A Qualitative Thematic Analysis Using the Common-Sense Model

**DOI:** 10.3390/ijerph20042920

**Published:** 2023-02-07

**Authors:** Cameron Little, Andrew P. Lavender, Cobie Starcevich, Christopher Mesagno, Tim Mitchell, Rodney Whiteley, Hanieh Bakhshayesh, Darren Beales

**Affiliations:** 1Curtin enAble Institute and Curtin School of Allied Health, Curtin University, Perth, WA 6102, Australia; 2Institute of Health and Wellbeing, Federation University Australia, Ballarat, VIC 3350, Australia; 3Institute for Health and Sport, Victoria University, Melbourne, VIC 3011, Australia; 4Pain Options, Perth, WA 6151, Australia; 5Aspetar Sports Medicine Hospital, Doha 29222, Qatar; 6Curtin School of Electrical Engineering, Computing and Mathematical Sciences, Curtin University, Perth, WA 6102, Australia

**Keywords:** anterior cruciate ligament, fear, qualitative, biopsychosocial, common-sense model (CSM)

## Abstract

Fear is a significant factor affecting successful return to sport following an anterior cruciate ligament (ACL) injury. However, there is a lack of understanding of the emotional drivers of fear and how fear beliefs are formed. This study qualitatively explored the contextual and emotional underpinnings of fear and how these beliefs were formed, with reference to the Common-Sense Model of Self-Regulation. Face-to-face online interviews were conducted with ACL-injured participants (*n* = 18, 72% female) with a mean age of 28 years (range 18–50 years). Participants were either 1 year post ACL reconstruction surgery (*n* = 16) or at least 1 year post injury without surgery (*n* = 2) and scored above average on a modified Tampa Scale of Kinesiophobia. Four participants were playing state-level sport or higher. Five themes emerged describing factors contributing to fear: ‘External messages’, ‘Difficulty of the ACL rehabilitation journey’, ‘Threat to identity and independence’, ‘Socioeconomic factors’, and ‘Ongoing psychological barriers’. A sixth theme, ‘Positive coping strategies’, provided insight into influences that could reduce fear and resolve negative behaviors. This study identified a broad range of contextual biopsychosocial factors which contribute to fear, supporting the notion that ACL injuries should not be treated through a purely physical lens. Furthermore, aligning the themes to the common-sense model provided a conceptual framework conveying the inter-related, emergent nature of the identified themes. The framework provides clinicians with a means to understanding fear after an ACL injury. This could guide assessment and patient education.

## 1. Introduction

The anterior cruciate ligament (ACL) provides mechanical stability and proprioceptive feedback within the knee. This is particularly important in sports requiring pivoting, change in direction, dynamic acceleration or deceleration [1]. Injuries to the anterior cruciate ligament (ACL) are common [2]. In Australia, nearly 200,000 ACL reconstructions were reported between July 2000 and June 2015 [3], with the incidence of ACL injuries increasing [4]. A systematic review of 69 articles reporting on 7556 participants with an ACL reconstruction found 45% of individuals did not return to competitive sport despite achieving successful surgical outcomes. Return to sport was affected by nonmodifiable factors including gender, age, and preinjury sports participation level, with elite athletes being more likely to return to previous levels of sport [5]. For those who do not return to sport, 65% report psychological barriers, with 77% of these people citing fear of reinjury as the primary issue [6]. Fear of reinjury may lead to behavioral manifestations such as muscle bracing and hesitancy in sport [7], maladaptive guarding or protecting the knee during movement [8], avoidance of certain movements or sports [9], hesitation to participate in rehabilitation, [9] and suboptimal performance [10]. Accordingly, research on the psychological impact of an ACL injury, especially fear, is a topic that has gained substantial interest in recent years [8,11,12,13]. Fear (defined as ‘a basic, intense emotion aroused by the detection of imminent threat’ [14]), and in particular fear of reinjury, appear to contribute to suboptimal outcomes after an ACL injury [11,12,15,16,17,18].

Previous qualitative studies have explored fear after an ACL injury, with a focus on the consequences of reinjury [19,20,21,22,23]. Themes such as a desire to avoid further surgery [21,22,23], nature of the preinjury sport imposing risk of reinjury [21], not wanting to undergo another long rehabilitation [21,22,23], social considerations such as loss of income or a reprioritization of sporting commitment [19,21,23], and concern over a decline in sporting competency [23] all contributed to fear. Fear of reinjury appears multifaceted and complex, with the exact construct underpinning this reaction being underexplored, with only one qualitative study reporting emotional responses [21]. The need for greater clarity around the construct of fear of reinjury may help improve our understanding of the barriers preventing people from achieving optimum outcomes following ACL injury [22]. 

The first aim of this study was to explore the contextual and emotional underpinnings of fear in people 1 year after ACL injury/surgery through qualitative examination. The second aim was to create a conceptual framework for how this cohort formed their injury beliefs by mapping the thematic results of the qualitative analysis to the Common-Sense Model of Self-Regulation (CSM). The CSM is a theoretical framework use to examine the thoughts, actions, and emotional responses to an injury or health threat, and from this context, assist in understanding how a belief system is formed [24]. The CSM has previously been used to aid in the interpretation of psychological factors within musculoskeletal disorders such as back pain [25], hip pain [26], and pelvic girdle pain [27]. Increased understanding of factors that underlie fear following ACL injury may help to identify athletes at risk of suboptimal return to sport and might assist targeted management strategies.

## 2. Materials and Methods

### 2.1. Study Design

A qualitative interpretive study design was used to explore the participants’ experiences of their ACL injuries. This design allowed the researchers to be guided by their clinical experience and knowledge in developing the interview schedule and in critically evaluating the data [28,29]. Thematic analysis checklist was used as a complimentary process to ensure trustworthiness of the analysis [30] (Appendix A). The study reporting was aligned to the Consolidated Criteria for Reporting Qualitative Research (COREQ) guidelines [31]. Ethics approval was granted by the Curtin University Human Research Ethics Committee (HRE2020-0655).

### 2.2. Participants 

Participants were recruited between October and December 2020 through a social media advertisement on Facebook. No financial compensation was offered. Non-probabilistic purposive sampling was used to ensure information-rich data were collected from ACL-injured individuals who could articulate their emotions and beliefs [32].

Inclusion criteria included people aged between 18–50 either 1 year post ACL reconstruction surgery or at least 1 year post injury without surgery [33]. Conservatively managed and surgically managed participants were combined in this sample due to similar long-term patient-reported outcomes for quality of life, activity levels, and knee function [34,35,36,37,38,39,40,41]. Participants had to report ‘above average’ fear, with a score of more than 16 on a modified Tampa Scale of Kinesiophobia (TSK) to be eligible (total score of a range of 0–51, with higher scores indicating higher fear, with a median kinesiophobia score of 17 being previously identified) [18]. All sporting levels and types of sports were included. Potential participants were excluded if they scored less than 17 on the TSK, had low back pain with radicular signs, had multiple pain sites, or came from a non-English-speaking background.

### 2.3. Participant Measures

Participant demographics included age, sex, year(s) ACL injury(s) occurred, whether they had an ACL reconstruction or not, and the TSK. The Anterior Cruciate Ligament–Return to Sport Index was used as a measure of psychological readiness to return to sport (scored between 0 and 100, with lower scores indicating higher impairment) [42]. Participants also reported on how much they had been ‘bothered’ by their knee in the last week across four constructs (all rated from 0–10, where 0 was no bother at all and 10 was extremely bothered): (1) fear of movement, (2) distress/anxiety related to knee movements, (3) lack of confidence, and 4) pain. Return to previous level of function was scored from 0 to 3, based on a return to sport consensus statement [43] (no return to any exercise (0), return to basic exercise (gym, etc.) but no sport (1), return to sport at a lower level than previous (2), or return to previous level of sport (3)).

Semistructured, one-to-one interviews were conducted using online video software (Microsoft Teams). A skeleton interview script was used to ensure a broad consistency of the interview questions, informed by the CSM (Appendix A). During these interviews, the focus was on fear, with an understanding that this may have considerable overlap with other psychological constructs [44]. Accordingly, participants were encouraged to drive the discussion in directions that particularly concerned them [45]. The interview process was iterative, whereby the interview questions evolved and changed over time. Interviews were conducted by a male physiotherapist, CL. CL has a post-graduate degree in sports physiotherapy with clinical experience managing ACL injuries in elite athletes. CL is also a PhD candidate with training in qualitative interviewing who had no prior relationship to the participants.

### 2.4. Sample Size

Sample size was determined by the point at which saturation of the qualitative data occurred; that is when no new themes were being discovered [46,47] and the themes that were generated from the reviewed information were deemed to be appropriate and adequate by the interdisciplinary team [47,48]. Saturation was deemed to be reached at 16 interviews. Two more interviews were then completed to ensure that no new themes were discovered.

### 2.5. Qualitative Data Analysis 

All interviews were audio-recorded, transcribed verbatim, and entered into NVIVO 12 (QSR International, Melbourne, Australia). Analysis followed an interpretive descriptive framework, a theoretically flexible framework allowing the construction of themes through subjective participant experiences [29].

Methodological steps for interpretive description in qualitative research were closely followed throughout the analysis [49]. This included independent coding of the data by two coders (CL and CS, a female physiotherapist with a post-graduate degree in musculoskeletal physiotherapy and a PhD candidate with experience in qualitative research). An inductive coding process was used, meaning codes were identified from the data and not preconceived. Coding was initially completed on the first twelve transcripts (out of eighteen). Coders then collaborated to discuss and synthesize their individual codes to create a code book. Coders then retrospectively applied the code book to all existing transcripts with a focus on identifying any quotes which could not be encapsulated by the existing codes, ensuring new codes would be created for the robustness of the code book. CL and CS reviewed codes and adjusted the code book to add new codes prior to presenting it to the broader interdisciplinary team of DB (a male specialist musculoskeletal physiotherapist and senior research fellow), CM (a male sport psychologist with extensive experience in fear and anxiety and 15 years of qualitative analysis experience), and AL (a male senior researcher and lecturer in neuromuscular function and accredited exercise scientist) for further discussion, negotiation, and refinement of themes. Four other transcripts were then coded, bringing the total to sixteen, with the goal to test, challenge, and further refine the code book. At 16 interviews, the code book was again discussed with the broader interdisciplinary team for further synthesis and refinement of final themes. Coders then independently coded another two interviews to ensure the emergence of no new themes. Final input was completed with the broader interdisciplinary team, CL, CS, DB, CM, AL, TM (a male specialist musculoskeletal physiotherapist), and RW (a male specialist sports physiotherapist), to further synthesize and ensure the credibility and trustworthiness of the overall themes, codes, and quotes. 

Completed themes were then mapped to the CSM. CL completed the initial mapping of themes to the CSM, partially guided by previous examples of musculoskeletal mapping [25,26,27]. The mapped model was then reviewed by some of the team (DB, AL, and CM), discussed, and refined, before being reviewed by the remainder of the team (CS, HB, TM, and RW). 

### 2.6. Community Engagement 

Community engagement was performed and reported according to the short form of the Guidance for Reporting of Patients and Public (GRIPP 2) [50]. The engagement group consisted of three people with previous ACL injuries (31-year-old male, 32-year-old female, and 33-year-old male). This engagement group provided input on the viability and usefulness of the project, on the interview schedule, time requirements, and on safety considerations. Trial interviews were completed with the engagement group prior to data collection, which allowed for further refinement of questions and interviewer training. Data from these trial interviews were not included in the full analysis. The consumer group was consulted for input during the analysis phase of the qualitative data and on the mapping of qualitative themes to the CSM.

## 3. Results

### 3.1. Participant Characteristics

A total of eighteen participants, all at least 1 year post ACL injury/surgery with above-average fear on the Tampa Scale of Kinesiophobia (TSK), were included (Figure 1). Information on the participants is detailed in Table 1. The average age was 29 years, 72% were female, with the average time since their last injury being 5.5 years. Sixteen of the eighteen had their injury surgically managed. Nine of the eighteen returned to some level of competitive sport, though only three to an equivalent level to that preinjury. Additional information about the nature and level of participants’ sports can be found in Appendix A.

### 3.2. Themes

Interviews lasted an average of 40.5 min (range 25 to 70 min). Full interview transcripts are provided as Appendix A. Five themes relating to fear were identified (supporting quotes can be found in Table 2, Table 3, Table 4, Table 5 and Table 6; text in quotation marks (Q) relates to participant quotes from this table). A sixth theme emerged related to positive coping strategies reducing fear (Table 7). 

#### 3.2.1. Theme 1: External Messages Driving Fear 

Participants reported they had received messages from external sources including friends and family, the internet (Q1.3), media (Q1.2) or healthcare professionals which heightened their fear. Participant 13 reported the surgeon saying it was ‘One of the worst injuries he had seen that year’ (Q1.5). These messages triggered significant emotional and behavioral reactions, such as Participant 11 who reported, ‘Since knowing it’s an ACL injury… I don’t go running’ (Q1.3), both contributing to and driving fear.

#### 3.2.2. Theme 2: Difficulty of the ACL Rehabilitation Journey Driving Fear 

Participants spoke about a fear of undergoing the rehabilitation journey again, with the psychological challenge, time commitments, and isolation all contributing to fear. Psychological challenges during rehabilitation were reported, including difficulties of a ‘set-back’ (Q2.1), motivational difficulties (Q2.2), and depressive episodes (Q2.5, Q2.6). They discussed the significant time commitment of ACL rehabilitation affecting their willingness to risk reinjury (Q2.11–3.12). Participants also remarked on the loneliness and isolation brought about by the rehabilitation process, such as Participant 2 who stated ‘The loneliness was every lift I ever did, every workout I ever did, every preinjury (exercise) you have got to do 90% of it on your own’ (Q2.7). The prior experience and potential to repeat rehabilitation was a significant contributor to fear in the cohort.

#### 3.2.3. Theme 3: Threat to Identity and Independence Driving Fear

Participants discussed a change in self-identity, sporting participation, and self-belief which contributed to fear. They spoke about losing their sense of athletic identity (Q3.1–Q3.5). Participant 1 stated, ‘That whole perception of me being the sporty person, yeah, it just kind of went out the window’ (Q3.1). Participants spoke about their diminished independence when they were injured or recovering from surgery (Q3.6–Q3.8). Participant 14 said, ‘The lack of independence … not being able to do things for myself and then relying on other people, that stresses me out’ (Q3.6). Or, Participant 7, ‘Having to ask someone to help you get up and go to the toilet or something like that. You just feel like a … burden’ (Q3.8). Within this cohort, the potential threat of further changes to identity and independence in the instance of a reinjury contributed to fear.

#### 3.2.4. Theme 4: Socio-Economic Factors Driving Fear 

Participants reported the economic and social implications of an ACL injury contributed to their overall fear of an ACL reinjury. The cost of both surgery and rehabilitation (Q4.3), as well as work and career setbacks, were discussed, such as Participant 3: ‘Oh, it left me pretty devastated… Losing my job…but then having to go and find another job, and all that stuff was pretty difficult and hard to take just because of a stupid soccer injury’ (Q4.4). Participants discussed social changes because of the injury, such as Participant 4: ‘Friendships changed. I just moved away from a lot of friends. I wasn’t able to drive initially. It’s changed my entire friendship group’. (Q4.5). Changes to socio-economic circumstances were seen to contribute to fear related to the previous ACL injury experience, as well as fear related to the potential of reinjury.

#### 3.2.5. Theme 5: Ongoing Psychological Barriers Driving Fear

Participants talked about their injury and the subsequent rehabilitation process resulting in ongoing psychological difficulties, such as anxiety, rumination, and avoidance activities, which heightened fear and contributed to avoidance of sporting activities. For example, anxiety, ‘My risk of re-tearing my knee is astronomical’ (Q5.3), rumination, ‘It’s constantly on my mind now’ (Q5.4), and avoidance behaviors, ‘Maybe I’ll just be a less active person’ (Q5.2). Participants reported traumatic memories from the point in time of the initial injury (Q5.9–Q5.13). They recalled their own injury experience in response to a trigger such as viewing an interview with an ACL-injured player (e.g., Q5.9), or in some cases, experienced spontaneously (e.g., Q5.10).

#### 3.2.6. Theme 6: Positive Coping Strategies Reducing Fear

Throughout the ACL injury recovery process, positive social, psychological, and self-management strategies were experienced which reduced fear and improved function. Social support from friends and family reduced isolation (Q6.1). Maintaining a strong connection with their sporting teams and organizations assisted with rehabilitation motivation and appeared to reduce the impact of negative changes to athletic identity (Q6.11). Self-empowerment and promotion of an internal locus of control around the injury appeared to reduce fear, such as Participant 11 who reported ‘After being diagnosed with a torn ACL, that’s when I started doing some research and I was like, ‘Okay, oh, this is pretty common, and this is where the ligament is, blah, blah, blah’ … Just that sense of not knowing that was all (that made) me distressed’ (Q6.15). Having a positive relationship with a healthcare professional could reduce the difficulty of rehabilitation, such as Participant 7, who reported, ‘If I didn’t have (such) a good relationship with my physios, I don’t think I would have had as good of an outcome’ (Q6.7).

### 3.3. Common-Sense Model

Mapping the themes to the modified Common-sense Model (Figure 2) provided a useful conceptual framework displaying the connectivity of the themes and how fearful beliefs may be formed after an ACL injury. Participants may have created an initial interpretation of their injury, influenced by external messages such as those from society and healthcare professionals (Theme 1). Participants could have then created an individualized representation of their injury, which may be influenced by the difficulty of the ACL rehabilitation journey (Theme 2), threat to identity and independence (Theme 3), and socioeconomic factors (Theme 4). This representation might then influence behavior, either negatively with psychological barriers, such as anxiety and avoidance (Theme 5), or lead to positive behaviors, such as gaining support from family and friends and conducting private research to reduce fear (Theme 6). Behaviors were likely being constantly appraised, with coinciding emotional reactions which could have further influenced participants’ representation of their injury as the CSM cycle continued its iterations.

### 3.4. Consumer Group Review of Results

After coding of the initial 12 transcripts, the consumer engagement group felt the themes were not broad enough or inclusive enough to capture the variety of participant responses, consistent with the research team’s belief that saturation had not been reached at that point. When presented with the final themes, the consumer group unanimously agreed that they covered the depth and breadth of participants’ personal experiences. The engagement group also provided feedback on the conceptual framework developed with mapping themes to the CSM (Figure 2). One reported he found it ‘oversimplified’. As a whole though, the consumer group reported that it was an interesting representation of their injury and helped to put into context how their beliefs about fear were formed. In response to the feedback, subthemes were expanded to add more depth to the model. Also included as Appendix A is a reflection of CL (first author, interviewer, and primary data analyzer) on how undertaking this study influenced his own beliefs and feelings on the topic.

## 4. Discussion

Fear of reinjury is a significant barrier to recovery following ACL injury [11,12,15,16,17,18]. The first aim of this study was to explore the contextual and emotional underpinnings of fear in people 1 year after ACL surgery/injury. Interviewing participants with above-average levels of fear based on the modified TSK, multiple themes emerged to help contextualize the experience of fear. These included external messages, ongoing psychological barriers, threat to identity and independence, difficulty of the ACL rehabilitation journey, and socio-economic factors (Table 2, Table 3, Table 4, Table 5 and Table 6). Broadly, these findings align to prior qualitative work in this area [19,20,21,22,23]. In addition, positive coping strategies that can reduce fear were identified. These included gaining strong support from family and friends, obtaining clear guidance on their rehabilitation program, maintaining a connection with sporting clubs or organizations, and conducting personal research on injury and rehabilitation best practices (Table 7). This aligns to previous research which explored strategies to reduce fear of reinjury and manage anxiety following ACL reconstruction [51]. The second aim was to create a conceptual framework for how this cohort formed their injury beliefs by mapping the thematic results of the qualitative analysis to the CSM. This process highlighted the connectivity of the themes and provided insight into formation of their fear beliefs (Figure 2). While fear has a specific focus in the ACL literature, the connectivity between themes suggests fear should be considered within a broader biopsychosocial context for an injured athlete.

### 4.1. External Messages

A role was identified for external messages from both healthcare professionals and society which contributed to fear (Theme 1). Within the low back pain literature, a range of information sources can affect an individual’s beliefs. For example, messages from healthcare professionals of damage and structural weakness encourage vigilance, worry, and guilt about movement [52]. The importance of messaging has not been seen in the previous ACL qualitative literature. However, it has been identified within broader sport-related knee injury research, wherein practitioner empathy and positive communication have been shown to benefit recovery [53]. Similarly, promoting positive emotional responses and a positive locus of control after an ACL injury can improve return to sport [54]. Positive messaging by physicians [55] and creating a supportive environment for injured athletes [56] can also positively influence recovery. These findings align with Theme 6. Indeed, messages that contributed to fear (Theme 1) were directly comparable to those that reduced fear (Theme 6).

### 4.2. Changes to Identity

Consistent with prior findings [22,57], participants spoke about changes to their identity in response to their ACL injury (Theme 4). Those with a high athletic identity may be more prone to suffer psychological hardships if injured, as they often derive their self-worth from their athletic capabilities. When self-worth is taken away, their sense of self can be threatened, and they may have a negative psychological reaction in response [6,58]. Interestingly, those with a high athletic identity are often very adherent to rehabilitation, which may improve return to sport [59]. Considering changes in athletic identity might be important for understanding fear. Those who identify strongly as an athlete may have more fear around returning to sport and could benefit from psychological intervention early in the rehabilitation journey.

### 4.3. Pain

Pain did not emerge as a significant contributor to fear. This is not to say that the appropriate management of pain is not important, as research has indicated that appropriate pain management assists ACL outcomes during early rehabilitation [19]. Within this cohort only a few participants recalled pain as a part of their ACL injury experience. These participants subsequently reported ongoing fear in the absence of pain, supporting the notion that pain is not a dominant feature in late-stage rehabilitation [60,61] and not a driving factor of fear after an ACL injury [19,20,21,22,23].

### 4.4. Common-Sense Model and Fear

Musculoskeletal injuries, including sport injuries, are complex biopsychosocial phenomena [62,63]. Recognizing the multidimensional nature of sports injuries is important, but it can be challenging to understand how contributing factors might interact. The CSM has been shown to have utility in providing conceptual frameworks for understanding the interaction of biopsychosocial factors in musculoskeletal disorders [25,26,27]. In those studies, the CSM has been used to explain how people might make sense of a pain disorder. The participants in this study differ in that pain is not a specific issue, and the onset of an ACL injury is a specific traumatic event. Like in pain disorders though, emotional responses [25], self-perception of illness [27], beliefs, and coping strategies [26] were all important in the CSM-based conceptual framework formulated here (Figure 1). The formation of fear and associated beliefs influence health behaviors, which in turn can strongly affect outcomes. For example, in those awaiting joint replacement surgery, a negative belief system is associated with a reduction in functional outcomes post surgery [64]. Conversely, a positive belief system after a total knee replacement is associated with improved function [65]. This is reflected in Theme 5 and Theme 6 of our results. Furthermore, how an injury is cognitively processed within people with osteoarthritis [66] and low back pain [67] can influence ongoing disability for up to six years. The way the subjects processed information in this study might well have long-lasting implications given ongoing higher levels of fear for up to 14 years after an ACL injury.

## 5. Clinical Implications

While clinical practice guidelines often acknowledge the importance of psychosocial issues for rehabilitation after an ACL injury, return to sport guidelines appears to be dominated by time and physical impairment criteria [32]. Rehabilitation focused on physical impairments in isolation does not correlate with optimal outcomes, including return to sport [17]. For those patients with above-average fear scores, there is a strong argument for an increased focus on fear in rehabilitation programs. The conceptual model presented here (Figure 1) provides guidance on how the CSM might be used to understand fear within a person-centered biopsychosocial approach. Assessment models of patient beliefs for musculoskeletal pain disorders might still be useful in guiding assessments within the return-to-sport phase post ACL injuries [68]. Recent qualitative research has provided insight into strategies used by competitive athletes to overcome fear of reinjury after an ACL injury [51]. Facilitating a strong athlete identity, building mental toughness, and integration of preparing the mind as well as the body for return to sport during rehabilitation were identified as positive strategies. 

## 6. Strengths, Limitations, and Future Research Considerations

This study employed a robust qualitative design, adhered to recommended reporting standards, and utilized a multidisciplinary team with a broad range of experiences to facilitate the trustworthiness of the findings. This included the use of a community engagement group and following a qualitative framework to inform all stages of the research. Recruitment and inclusion criterion allowed for a broad range of participants to be recruited from the community (rather than from a specific surgeon, age or sporting club). These participants had a varied number of injuries and times since their ACL injury/surgery. As this sample was mostly females (13/18), this perhaps does not reflect previous epidemiological data, which indicate that between 2000–2015, 68% of ACL reconstructions in Australia were on males [3]. It is the opinion of the research group that this does not affect the generalizability of the results. A potential limitation was the inclusion of people defined as having ‘above-average fear’ on the modified TSK [18]. The psychometric properties of the TSK have been investigated within other conditions, such as chronic low back pain and fibromyalgia [69], however, the nature of an ACL injury differs from chronic pain. While the modified TSK has been previously used in ACL research, there is some indication that it is valid only in the early-to-middle phase of rehabilitation (up to 6 months) and not in the late phase of rehabilitation [60].

## 7. Conclusions

This study identified a broad range of contextual factors that underpinned fear within people who are past 1 year after an ACL injury/surgery. This lends support to the notion that an ACL injury should not be viewed through a purely physical lens. Rather, a complex interplay of biopsychosocial factors throughout the injury journey may contribute to fear and drive behaviors. Consideration of fear in isolation from the broader biopsychosocial context might be detrimental to understand the needs of any specific athlete. Using the CSM to conceptualize the role of fear in conjunction with other biopsychosocial factors could assist clinicians to better identify and manage fear as part of a comprehensive treatment approach.

## Figures and Tables

**Figure 1 ijerph-20-02920-f001:**
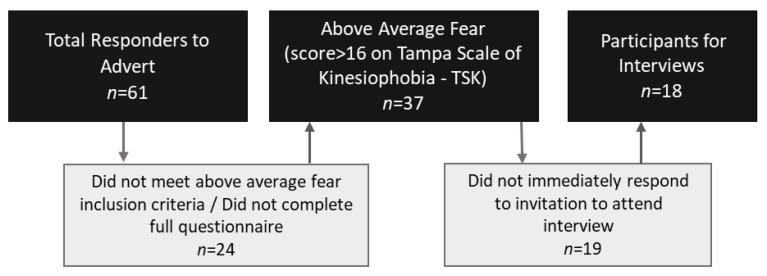
Flow diagram for participant recruitment.

**Figure 2 ijerph-20-02920-f002:**
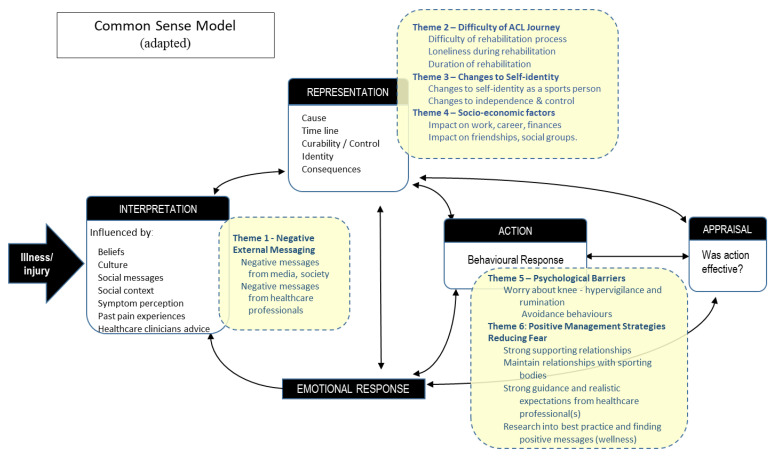
Common-Sense Model (based on the work of Leventhal—various papers 1965–2016). Themes and subthemes in subjects with above-average fear at least one year after an ACL injury, mapped to the Common-Sense Model.

**Table 1 ijerph-20-02920-t001:** Individual Participant Characteristics.

ID	P1	P2	P3	P4	P5	P6	P7	P8	P9	P10	P11	P12	P13	P14	P15	P16	P17	P18	Mean (SD)	Median (IQR)
Age	22	25	33	28	24	22	28	32	23	44	31	33	32	31	24	23	36	25	28.0 (5.86)	28.0 (23.8, 32.3)
Sex	F	F	M	F	F	F	M	M	F	F	F	F	F	F	M	M	F	F	M:5, F:13	
Tampa Scale of Kinesiophobia ^1^	19	17	33	23	23	18	21	17	17	21	31	25	17	22	25	30	18	30	22.6 (5.35)	21.5 (17.8, 26.3)
Anterior Cruciate Ligament Return to Sport Index ^2^	39.4	34.2	25.8	41.7	52.5	28.3	53.3	63.3	66.7	66.7	20.0	16.7	65.8	18.3	31.7	47.5	41.7	15.8	40.5 (17.93)	40.6 (24.4, 55.8)
Fear of movement ^3^	1	4	3	0	4	2	0	0	0	3	10	0	0	6	3	8	2	2	2.67 (2.91)	2.0 (0, 4)
Distress/Anxiety related to knee movements ^3^	1	1	5	1	4	0	0	0	0	3	8	5	0	6	4	7	3	1	2.72 (2.65)	2.0 (0, 5)
Lack of confidence in knee movements ^3^	0	8	7	2	5	2	0	0	0	7	10	5	0	6	4	6	5	4	3.94 (3.15)	4.5 (0.0, 6.25)
Pain ^3^	1	7	2	1	6	0	2	0	0	6	3	0	0	6	1	2	3	1	2.28 (2.4)	1.5 (0.0, 3.8)
Surgery	Y	Y	Y	Y	Y	Y	Y	Y	Y	Y	N	Y	Y	Y	Y	Y	Y	N	Y:16, N:2	
Psychological support provided	N	Y	N	N	N	N	N	Y	N	N	N	N	N	N	N	N	N	N	Y:2, N:16	
Return to previous level of function ^4^	2	2	0	2	3	2	3	1	3	1	2	1	1	1	2	1	1	1	Count ^5^: 0:1 1:8 2:6 3:3	
Year(s) of ACL injury ^6^	’16	’17 ’18	‘16	’16	’17	’11 ’16	‘17	’11 ‘12	‘15	‘18	‘15	‘08	‘08	’06 ‘09 ’13	‘19	‘19	‘06	‘19		

^1^ Tampa Scale of Kinesiophobia. This scale is rated 0–51, with a higher score indicating higher levels of fear. ^2^ Anterior Cruciate Ligament Return to Sport Index. This scale is rated out of 100, with lower scores indicating higher impairment. ^3^ ‘In relation to your knee, in the last week how much have you been bothered by …?’ (rated 0–10, 0 indicating not bothered at all, 10 indicating extremely bothered). ^4^ Return to previous level of function: scoring: 0 = no return to any exercise, 1 = return to basic exercise (gym, etc.) but no or minimal sport (i.e., noncompetitive), 2 = return to sport at a lower competition level than previous, and 3 = return to previous level of sport. ^5^ 0:1 refers to one person scoring zero. 1:8, refers to 8 people scoring 1, etc. ^6^ What year the ACL injury occurred in., e.g., ‘18 refers to the year 2018.

**Table 2 ijerph-20-02920-t002:** Supporting quotes to theme 1—external messages driving fear.

**Messages from society**	
Q1.1	If you just hear ‘ACL’ it kicks in that fear like, ‘wow, this is bad’. (P9)
Q1.2	… seeing idols and professional athletes, the people that you look up to on a day-to-day basis, seeing them with ACL injuries and then just hearing their stories that they can’t go back and perform at a level which they used to (P14)
Q1.3	Since knowing that it’s an ACL injury … I don’t go running. (P11)
**Messages from healthcare professionals**	
Q1.4	He said (the surgeon) to take up golf. He was really angry about the last one (referring to a rerupture of surgically repaired ligament) because it ruined all of his beautiful work. (P14)
Q1.5	He said (the surgeon) it was ‘One of the worst injuries he had seen that year’ and he said, ‘you must have been in a world of pain at the time’ (P13)

**Table 3 ijerph-20-02920-t003:** Supporting quotes to theme 2—difficulty of the ACL rehabilitation journey driving fear.

**The difficulty of the rehabilitation process and its psychological effects**	
Q2.1	Any time that I did feel anything that twitched the knee, or it buckled or it didn’t feel right in that first six months, it was almost as detrimental mentally than it did the knee. I was, like, ‘Whoa, I’ve got a long way to go’. every time, every time’. (P14)
Q2.2	… the rehab just felt like a chore that I had to do rather than something that I wasn’t really invested in…I wasn’t really great at that … (P10)
Q2.3	when you tear your ACL, you’ve got yourself to deal with and you’ve got to push yourself through it all (P15)
Q2.4	I remember waking up from surgery, and it hurt and I was just like, “Oh, why did I do this?” This is really painful now. That was really annoying. I second-guessed why I got the surgery done. (P6)
Q2.5	When I was in the Zimmer splint for my friend’s wedding, we were quite high up and I just kind of had, like, a quick thought of, what if I jumped—it got pretty dark. (P14)
Q2.6	I didn’t have a great time when I was in the splint, I pretty much was just at home in bed for that whole month and couldn’t walk around much. That was very miserable. I was like, ‘I would really hate to have to do all that again’. (P5)
**Loneliness during rehabilitation**	
Q2.7	The loneliness was every lift I ever did, every workout I ever did, every preinjury (exercise) … you have got to do 90% of it on your own. (P2)
Q2.8	It was very isolating and it’s just so monotonous. You don’t feel anything and that sucks. (P5)
Q2.9	you know they’re in for that long recovery and going back to that like loneliness… it’s like your road to recovery … you just feel lonely through the whole process (P1)
Q2.10	You go from a hospital bed, everyone’s there for you, and then suddenly, everyone has to go to work… and everything is by yourself. (P2)
**The duration of the rehabilitation and recovery process**	
Q2.11	I’ll be like crying too some nights. Like is it ever going to end or whatever (P1)
Q2.12	An ACL sucks up a good six to 12 months … minimum. In whatever plans I might have for getting myself together, you might as well write off 12 months of that. I don’t have enough 12 months left while I’m motivated and I’m feeling youngish (P10)
Q2.13	there was a period of time within my recovery… and I just was really over it and I remember that’s when I was quite disheartened (P9)

**Table 4 ijerph-20-02920-t004:** Supporting quotes to theme 3—threat to identity and independence driving fear.

**Changes to self-identity as an athlete/sports person/competitor**	
Q3.1	I wasn’t able to be that sporty person and always be out training, playing sport … That whole perception of me being the sporty person, yeah, it just kind of went out the window. (P1)
Q3.2	It was just one of those things where I was now the knee person, the person with the knee injury. I used to be the person that would say yes to anything. I would just say, ‘Yes, let’s do it, give it a try’. Now I don’t. (P12)
Q3.3	I can’t play basketball anymore because of my knees, and I’m like, I’m 25 … it’s disabling a little bit. Physically, I can do those things, but it’s the mental factor that’s stopping me. It’s frustrating, the fear is what’s stopping me. (P2)
Q3.4	I was very competitive at hockey, I was playing a good grade, and I just wouldn’t be able to get back to that (P4)
Q3.5	Everyone said I should have been right to play, so I did go back to try and play. I was just like, ‘No, I can’t. There’s something wrong, so I can’t keep playing’. … I just never went and tried playing again (P12)
**Changes to independence**	
Q3.6	The lack of Independence… not being able to do things for myself and then relying on other people, that stresses me out (P14).
Q3.7	it’s quite humiliating to have to get on the tram in crutches, that sense of I want to be independent in that sense with my own movement and it’s that sense of uselessness (P18)
Q3.8	Having to ask someone to help you get up and go to the toilet or something like that. You just feel like a … burden. (P7)

**Table 5 ijerph-20-02920-t005:** Supporting Quotes to theme 4—socio-economic factors driving fear.

**Impact on work, career, and/or financial situation**	
Q4.1	Wow. I wasn’t able to work for basically nine months. I was pretty new in my career at that point in time, so there went work (P4)
Q4.2	My Mum is very watchful of money, so if I was doing something that maybe she thought could hurt me again, she would be like, ‘You don’t want to have to pay for a surgery and blah, blah, blah’. If I were to say to my Mum I was going to play lacrosse again, she would be like, ‘I don’t think that’s a good idea’. (P18)
Q4.3	Well, I have a very expensive left knee. I don’t want to make it more expensive than it currently is. I think I’ve spent enough money on it (P6)
Q4.4	Oh, it left me pretty devastated. Losing my job…but then having to go and find another job and all that stuff was pretty difficult and hard to take just because of a stupid soccer injury. (P3)
**Impact on friendships and social grouping**	
Q4.5	Friendships changed. I just moved away from a lot of friends. I wasn’t able to drive initially. It’s changed my entire friendship group (P4)
Q4.6	I would have liked to have taken up more you know activities, you know dance or you know something different, but I just feel like I can’t because of my knee (P14)

**Table 6 ijerph-20-02920-t006:** Supporting Quotes to theme 5—ongoing psychological barriers driving fear.

**Anxiety, rumination, and avoidance**	
Q5.1	It makes me anxious as well, like in the future…if I get pregnant… is my knee is going to hold up during that time as well. (P14)
Q5.2	In your brain, you can go, ‘Oh, if you’re a really active person, maybe really active people are more likely to do an ACL twice. Maybe I’ll just be a less active person, then I’ll avoid that possibility’. (P10)
Q5.3	My risk of re-tearing my knee is astronomical… I think about skiing and I go, ‘Woah, no’, but I would love to ski. (P2)
Q5.4	It’s constantly on my mind now in everyday life whether that’s walking, running or standing up. (P15)
Q5.5	I think the fear ultimately stems from doing it again, the pain of that, the cost of that and the time that will take from my life again. I just don’t want to be a victim to it again. (P18)
Q5.6	Constant analysis … instead of being able to go out and 100% enjoy something, you spend 50%, 75% of the time thinking or analyzing your knee … If I’ve got any unexplained pain in my knee that I can’t explain away … then I’m really afraid to do anything other than just walking. (P8)
Q5.7	After I had done all my rehab, if people… frightened me or got too close, I actually started having what I could call panic attacks, like hyperventilating and just shutting down. (P5)
Q5.8	It’s constantly on my mind now in everyday life whether that’s walking, running or standing up. (P15)
**Trauma associated with the initial injury**	
Q5.9	… from watching AFL… hear them say, oh I felt the pop, and it sends shivers down your spine … it’s so easy to do, it looks so easy. I was scared to go back so I never … I still think about it because it just triggers those memories and stuff again. (P1)
Q5.10	I used to have flashbacks … I’d just be driving or walking or anything and I’d just have these flashbacks to it happening. (P2)
Q5.11	Probably the most painful thing that’s ever happened to me. I remember when it happened, I just physically collapsed … I could’ve been shot by a bullet, I could’ve been hit by a bus, I had no idea. (P18)
Q5.12	I know the feeling of the pop as well, it’s like I can still imagine it to this day … that’s had such an effect on my life. (P1)
Q5.13	It was very painful and excruciating. I remember every part of it. Everything that I did that was cringy during it because I felt my knee twist, I felt a pop. I screamed, ‘I heard a pop’. (P16)

**Table 7 ijerph-20-02920-t007:** Supporting Quotes: Theme 6—Positive coping strategies reducing fear.

**Gain strong support from family and friends**	
Q6.1	… Good family and friends’ support to begin with. The best thing for me is that one of my very good friends he did his ACL about a week after me. We ended up being able to, I guess just talk about it. Being each other’s own little support network (P4)
Q6.2	Previously I’d had to get my boyfriend to hold me up on the bike, try and pedal and it’s like in my mind I couldn’t do it. Then as soon as he (the surgeon) told me that (I could ride) the next morning I said, ‘I’m going to go try and ride around the block’. It was like I could just do it. (P18)
Q6.4	It’s nice to relate to someone about an injury and talk about things that you might be feeling with it not feeling 100% (P13)
**Obtain clear guidance on rehabilitation program**	
Q6.7	If I didn’t have (such) a good relationship with my physios, I don’t think I would have had as good of an outcome. (P7)
Q6.8	The owners of the gym, they helped me accept it in that they validated that it is an injury and it’s okay that it has a big impact on my life. Just because other people might be going through really, really bad things, my life experiences are different to the next person. If this is the worst thing that ever happened to me, that’s okay. (P4)
Q6.9	My physio was fantastic and he always, yeah, gave me a lot of reassurance (P9)
Q6.10	My physios and the doc and my surgeon were really good at outlining … the stages and the roadmap of how we’re going to get back…and always available to answer questions (P7)
**Maintain connection with sporting clubs or organizations**	
Q6.11	Once I could start walking around my Frisbee team gave me an assistant coach position or a training coach position and so that meant I could start going down to trainings and watching and try help coach and learn things from that perspective. I think that that’s when my rehab and getting back into sport really picked up. (P5)
Q6.12	People knew that you’re always coming back and so it was, “Just do as much as you can no one’s going to judge you for it. We know that you are worthy of being on this team no matter what level you’re playing at”. (P5)
**Conduct private research on injury and rehabilitation best practice**	
Q6.13	I’d read up with athletes and a lot of athletes go through very public rehabs these days, so you have some kind of idea and understanding of what it’s going to be like. (P7)
Q6.14	I was gathering information from people in the (online) community who had torn their ACLs before…Finding out how to go through it the best way is something that gave me a lot of peace (P16)
Q6.15	After being diagnosed with a torn ACL, that’s when I started doing some research and I was like, “Okay, Oh, this is pretty common, and this is where the ligament is, blah, blah, blah” … Just that sense of not knowing that was all (that made) me distressed. (P11)

## Data Availability

The data presented in this study are available on request from the corresponding author. The data are not publicly available due to privacy reasons.

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
