# Peer review of "Understanding Fear after an Anterior Cruciate Ligament Injury: A Qualitative Thematic Analysis Using the Common-Sense Model"

_ijerph, 2023, doi:10.3390/ijerph20042920_

Round 1
Reviewer 1 Report
This is a well written detailed manuscript that is ready for publication after very minor revisions. It takes and innovative approach to assessment and explaining the factors that impact return to sport after an injury. It has scientific merit. The significance of returning to the sport however has to be explicated in the beginning of the manuscript.
Content
Abstract: This sentence is too long and too much explanation for an abstract. Please condense it to the main point. “Aligning the themes to the Common-Sense Model provided a conceptual framework that conveyed the inter-related nature (connectivity) of the identified themes, and how these themes might influence ACL injury related beliefs. The conceptual framework provides clinicians with means to understanding long-term fear after an ACL injury. This could guide assessment and patient education.”
Ex. The Common-Sense Model provided a framework to organize the themes that can be used to guide patient assessment and education.
Study significance. The significance of individuals not returning to the sport needs to be explicated. How could this on a grand scale impact the sport and what negative effects on a grander scale will result when people do not return to the sport.
Mechanics: no problems
Organization: no problems
Reviewer 2 Report
In the title I would add "a thematic analysis"
20 methodologically, I would recommend first presenting the inclusion criteria.. the type of population you wanted to reach.. afterwards I would suggest putting in the abstract the male/female ratio, average age and practiced sport (especially if professionals or amateurs)
28 This conclusion is intriguing, but I would suggest providing conclusions in light of your findings in the abstract
36 As mentioned earlier, I would suggest dividing amateurs and professionals. The RTP in professionals is almost essential, even if only hypothetically, in amateurs it is underestimated and little studied (it might be a rationale of your study)
50 In this regard, I would also underline the type of sport practised...because skiing changes the practitioner's perception of the risk context
81 experience level? Sports practiced?
Regarding the methods I suggest to evaluate the Braun and Clarke checklist
Table 1 is not obvious, but complete and intelligible, therefore I suggest greater rigor of eligibility (in methods section)
187 methods statements.. in the results, could I suggest providing the mean duration of an interview?
If you have previously set the interview themes, then define it in the methods section with references and motivations
By convention I recommend placing the limitation paragraph as the last section of the discussion, with respect to clinical impact
401 suggested a broad range of contextual
Round 2
Reviewer 2 Report
Dear authors, this thematic analysis has been reinforced with greater methodological accuracy. has a small sample, but still I can suggest the suitability for publication of your paper.